# Supplementary Tryptophan Fed to Sows Prior to and after Farrowing to Improve Piglet Growth and Survival

**DOI:** 10.3390/ani11092540

**Published:** 2021-08-30

**Authors:** Amy L. Munn, Alice C. Weaver, William H. E. J. van Wettere

**Affiliations:** 1Davies Livestock Research Centre, School of Animal and Veterinary Sciences, The University of Adelaide, Roseworthy, SA 5371, Australia; amy.munn@adelaide.edu.au; 2Turretfield Research Centre, South Australian Research Development Institute, Rosedale, SA 5350, Australia; alice.weaver@sa.gov.au

**Keywords:** calcium, melatonin, viability

## Abstract

**Simple Summary:**

Pre-weaning mortality is a significant economic and welfare issue for the Australian pig industry. Tryptophan can increase serotonin and melatonin production. Serotonin can decrease stress and indirectly increase calcium, which may improve sow health. Meanwhile, melatonin may decrease stillbirths and improve piglet viability at birth and, in turn, increase survival to weaning. This study determined whether feeding 0.16%, 0.42% or 0.56% tryptophan (per kg of feed) to sows during late pregnancy until seven days of lactation could improve piglet survival and viability to weaning and increase the levels of calcium and melatonin in sows. Supplementing tryptophan at levels of 0.42 and 0.56% increased piglet survival compared to no supplementation but did not have an effect on piglet viability. Furthermore, tryptophan supplementation did not increase sow melatonin and calcium levels compared to 0.16%. Further research is required to understand how tryptophan may improve piglet survival, particularly through sow maternal behaviour, and if 5-hydroxytryptophan (the form of tryptophan that directly converts to serotonin and melatonin) would further improve piglet survival.

**Abstract:**

Tryptophan indirectly increases plasma calcium levels, which may improve sow health, and melatonin production, which may improve piglet survival when supplemented during late gestation and lactation. It was hypothesised that tryptophan would increase piglet survival and increase sow circulating melatonin and calcium. Seventy-two multiparous (Landrace x Large White) sows were allocated to either control (0.16% tryptophan; *n* = 24), low tryptophan (0.42%; *n* = 24) or high tryptophan (0.56%; *n* = 24). Piglet viability measures consisted of weights, behaviour, meconium staining, rectal temperature, blood glucose and serum immunoglobulin G concentration. Blood samples collected from sows were analysed for melatonin (two daytime and three night-time samples; *n* = 17) and calcium (two samples pre- and post-farrowing; *n* = 14). Both tryptophan treatments increased piglet survival compared to the control group (*p* < 0.001). Tryptophan had no effect on piglet viability (*p* > 0.05) and no effect on sow plasma melatonin and calcium concentrations compared with the control group (*p* > 0.05) except at 21:00 when low tryptophan sows had higher melatonin concentration compared with high tryptophan (*p* = 0.011). Further research to understand the mediating effects of tryptophan (particularly 5-hydroxytryptophan) on piglet survival, including sow behaviour, is warranted.

## 1. Introduction

In Australia, approximately 18% of piglets die between the start of parturition and weaning [1], resulting in an annual loss and industry cost of 1.2 million piglets and AUD 60 million, respectively [2,3]. Of these mortalities, death during parturition is the leading cause of pre-weaning mortality. Approximately 5–8% of piglets are stillborn, with the primary cause being intra-partum hypoxia, typically due to a prolonged parturition duration or being born later in the birth order [4,5]. Energy reserves and viability are reduced in hypoxic piglets which survive parturition, as well as those with low birth weight. These piglets are more vulnerable to other leading causes of mortality, such as crushing by the sow and starvation within the first three days of life [6]. Therefore, to improve survival at birth and weaning, a possible solution is to increase oxygen availability and nutrient supply to the fetuses prior to parturition through sow nutrition [7,8]. 

Tryptophan supplementation in late gestation may benefit piglet growth and survival; however, minimal research is available and has varying results. According to the NRC requirements of swine, the tryptophan requirement of lactating sows is 0.18 and 0.17% per kg of feed for primiparous and multiparous sows, respectively [9]. Paulicks et al. (2006)*,* Fan et al. (2016)*,* Miao et al. (2019) and Condous and van Wettere (unpublished) demonstrated that total tryptophan levels of 0.26, 0.28, 0.32 and 0.60% per kg of feed, respectively, increased piglet growth and/or survival when supplemented 6–11 days (at farrowing house entry) prior to parturition and throughout lactation [10,11,12]. However, Xu et al. (2018) found that tryptophan levels of 0.39% per kg of feed negatively impacted piglet survival at birth [13]. Despite this, the metabolites of tryptophan, serotonin and melatonin, appear to provide benefits related to piglet viability and may explain the benefits seen in piglet growth and survival, as discussed below.

Tryptophan, through hydroxylation, is converted to serotonin which is a neurotransmitter involved in the regulation of appetite and mood [14,15]. An increased level of tryptophan (0.32% per kg of feed) increased sow feed intake and milk output, resulting in higher average daily weight gain of the litter [12]. Cortisol levels 24 h post-parturition was 94% lower in sows fed a diet containing a tryptophan level of 0.6% per kg of feed compared with 0.2% per kg of feed (Condous and van Wettere, unpublished). In the same study, the 0.6% tryptophan diet altered sow behaviour post-parturition, resulting in fewer posture changes and less time standing which tended to reduce piglet overlays. In addition, the weaning weight of low birth weight piglets (<1.1 kg) was increased by tryptophan supplementation. In addition to appetite regulation and mood, serotonin is involved in the regulation of calcium mobilisation. Although calcium mobilisation may not provide benefits to sow offspring directly, maintaining an adequate level of calcium at the start of lactation is important for both sow health and welfare. Increasing milk production can lead to unexplained sow deaths and is speculated to be associated with hypocalcaemia related disorders [16]. In cattle, tryptophan supplementation (1 mg/kg body weight (BW)) increased calcium plasma levels and prevented hypocalcaemia [17]. Therefore, improving sow performance by increasing levels of tryptophan supplementation, mediated by the effects of serotonin, may increase piglet viability post-birth. 

In addition to serotonin, melatonin production may also increase when sows are supplemented with tryptophan. In sheep, tryptophan supplementation (222 mg/kg BW) increased melatonin production compared to no supplementation [18]. Melatonin has been shown to decrease oxidative stress, which indirectly increases umbilical blood flow, increasing nutrient and oxygen supply to the developing fetus. When melatonin was fed to pregnant ewes, the effects of artificially induced hypoxia in lambs was reduced, and fetal weight was increased [19,20]. If tryptophan supplementation can increase melatonin in gestating sows, it may ameliorate the negative effects of hypoxia in piglets, resulting in fewer stillbirths. Higher melatonin levels may also increase fetal energy reserves through an increase in nutrient and oxygen supply in utero. As a result, piglet vigour will be increased, leading to more rapid colostrum ingestion post-birth, greater mobility and reduced risk of crushing by the sow.

Most maternal tryptophan supplementation studies in sows have focussed on sow performance indices such as lactational weight loss and milk yield. Although benefits for piglet survival and weights have been shown [10,11,21,22,23], the impact of tryptophan on piglet viability has yet to be established. Furthermore, increasing levels of calcium and melatonin through tryptophan supplementation has only been studied in cattle, sheep and rats, respectively, and have not been established in sows [17,18,24]. Therefore, the aims of the present experiment were to; (1) determine if supplementing sow diets with tryptophan would increase piglet growth, vitality and survival to weaning, and (2) determine whether tryptophan supplementation would increase circulating melatonin and calcium in the sow. It was hypothesised that tryptophan supplementation would increase piglet viability and survival and increase melatonin and calcium levels in the sow. 

## 2. Materials and Methods

This experiment was approved by the University of Adelaide’s Animal Ethics Committee (S-2019-077) and conducted in accordance with the ‘Australian code for the care and use of animals for scientific purposes 8th edition’ (National Health and Medical Research Council: Canberra, 2013). All animal work was conducted at the University of Adelaide’s piggery, Roseworthy, South Australia.

### 2.1. Animals, Housing and Treatments 

A total of 72 multiparous Landrace x Large White sows (parities 1–6 prior to farrowing) were used in this study. Sows were mated via artificial insemination with a terminal sire semen mix. Sows were randomly stratified according to weight and parity to one of the following three treatment groups; (1) control, basal diet (containing a tryptophan level of 1.6 g/kg of feed (0.16%) with no additional tryptophan supplementation (*n* = 24); (2) low tryptophan (Low), basal diet with tryptophan supplementation of 2.6 g/kg feed (0.26%; total tryptophan = 4.2 g/kg feed; *n* = 24); (3) high tryptophan (High), basal diet with tryptophan supplementation of 4 g/kg feed (0.4%; total tryptophan level = 5.6 g/kg feed; *n* = 24). The trial was conducted over three farrowing replicates from November 2019 until January 2020 (late spring to mid-summer). Sows were group-housed from mating until entry to the farrowing house at approximately day 107.7 ± 0.2 of gestation. Sows were housed in conventional farrowing crates until weaning (19.4 ± 0.2 days post-parturition) with ad libitum access to fresh drinking water. A heat mat was provided in each farrowing crate for an additional heat source for the piglets. Where possible, cross-fostering occurred within treatments. Cross-fostering was minimised if litter size exceeded the number of available teats or if a litter had high birthweight variation and low birthweight piglets were unable to compete for teats. Additional piglets from outside the experiment were fostered onto sows in the experiment only if necessary and if teat capacity allowed (these piglets (*n* = 15) were excluded from the piglet analysis). All sows received a commercial lactation diet (14.2 MJ DE/kg as fed, 16.88% Crude Protein, 5.52% Crude Fibre; Ridley Corporation Limited, Wasleys, South Australia) upon entry to the farrowing house until weaning. Tryptophan supplementation began the day after farrowing house entry at day 108.0 ± 0.2 of gestation and ceased 7 days post-parturition (Table 1). Sows were fed twice daily at 07:00 and 15:00, receiving 2.5 kg feed per day in the lead up to parturition. For the treatment groups receiving supplementary tryptophan, tryptophan powder (Alltech Lienert, Roseworthy, SA, Australia) was top-dressed at each feeding time. A step-up feeding program was used during lactation. 

### 2.2. Animal Measures

#### 2.2.1. Sow Measures

Upon entry to the farrowing house, and on days 1.4 ± 0.1 and 19.0 ± 0.2 of lactation, sows were weighed, and the backfat at the P2 position was measured using an ultrasound machine and a 7.5 mHz linear probe (Esaote, Genova, Italy). A colostrum sample (5–10 mL) was collected across all teats following the birth of the first piglet. Colostrum was immediately analysed upon collection for total solid content (%) using a digital hand-held refractometer (Starr Instruments: Model DBR-1). Colostrum was then divided into three aliquots and frozen at −20 °C until further analysis.

#### 2.2.2. Farrowing Measures

Sows that were farrowed during the day were supervised to determine piglet birth order, inter-piglet birth intervals (min), total farrowing duration (min) and the number of piglets born alive, dead or mummified. If an inter-piglet interval exceeded 90 min, an internal examination was performed. Each piglet was ear-tagged at birth for individual identification. The time taken for the piglet to stand and then successfully attach to a teat and suck was recorded. If a piglet failed to accomplish these behaviours within 3 h, the piglet was either placed on the heat mat or placed on a teat. Finally, the degree of meconium staining was recorded to indirectly measure birth trauma [25] using the following scale: 1 = no staining, 2 = light staining, 3 = moderate staining and 4 = severe staining.

#### 2.2.3. Piglet Measures and Blood Collection

Piglets were individually weighed and sexed 4 h after the birth of the last piglet of the litter and weighed again at 24 h, 3 d, 7 d and 17 d of age. Rectal temperature was measured at 4 h and 24 h of age to measure the thermoregulation ability of the piglet. For all litters with known birth order, a blood sample was collected at 24 h of age from the first two, middle two and last two piglets born. The blood sample was collected from the jugular vein using a 21 G 1” needle and 3 mL syringe. Blood glucose concentration was measured immediately (Accu-Chek Performa, Sydney, New South Wales, Australia), and samples were dispensed into a 4 mL clot activator vacutainer tube (BD Vacutainer, BD, Belliver Industrial Estate, Plymouth, UK) and stored at 4 °C for approximately 24 h. The samples were then centrifuged at 1512× *g* for 10 min, and serum was divided across two aliquots and frozen at −20 °C until further analysis. Any piglet mortalities were recorded (date and cause of death), and piglets were euthanised if any deformities negatively impacted the quality of life.

#### 2.2.4. Sow Melatonin and Calcium

A subset of sows was randomly selected to measure plasma melatonin (*n* = 18) and calcium (*n* = 17). Sow blood samples were collected via jugular venipuncture using an 18 G 1.5” vacutainer needle and 9 mL Lithium Heparin vacutainer tubes (BD Vacutainer, BD, Belliver Industrial Estate, Plymouth, UK). For the melatonin samples, five blood samples were collected over a 24 h period that included two daytime samples (11:00 and 16:00) and three night-time samples (21:00, 01:00 and 05:00) (methods adapted from [26]). Melatonin blood samples were taken at day 11.2 ± 0.2 of tryptophan supplementation (4.9 ± 0.2 day of lactation). Blood samples for calcium analysis were collected on day 113 and 114 of gestation and day 1 and day 2 of lactation at approximately 10:00, with the assumption of farrowing occurring on day 115 of gestation. Samples were centrifuged at 1512× *g* within 1 h of collection for 10 min. Plasma was collected from each sample, divided into three aliquots and stored at −20 °C until further analysis. Plasma samples were analysed at The University of Adelaide Assay Research Facility. Blood plasma melatonin concentration was analysed via reverse-phase C-18 column extraction of 500 µL of plasma, followed by double-antibody radioimmunoassay (RKMEL-2, Buhlmann Laboratories AG, Schönenbuch, Switzerland). The assay was based on a previously validated method using Kennaway G280 anti-melatonin antibody [27] and [125I]2-iodomelatonin as the radioligand following the protocol provided by Buhlmann Laboratories. For optimal determination, samples were diluted at 1:4, making the lowest limit of quantitation of the assay 2.0 pg/mL. Samples were assayed in duplicate, and the intra-assay coefficient of variation of the assays was 7.3%. The inter-assay coefficient of variation of the assay at the low concentration quality control level was 2.8%, and the inter-assay coefficient of variation at the high concentration quality control was 14.1%. Blood plasma calcium was analysed via colourimetric assay (MAK022, Sigma Aldrich, St. Louis, MO, USA), which utilises the chromogenic complex formed between calcium ions and o-cresolphthalein. Each 10 µL sample was diluted 1:5 in ultrapure MilliQ water (Merck, Darmstadt, Germany) to ensure determinations remained within the assay standard curve of 1.0–5.0 nmol/µL. Samples were analysed in duplicate, and the intra-assay coefficient of variation of the assay was 5.2%.

### 2.3. Piglet Serum and Colostrum Immunoglobulin G, Total Solid/Protein Analysis

Piglet serum and sow colostrum were analysed to determine immunoglobulin G (IgG) concentration by a previously validated radial-immunodiffusion assay developed by the University of Adelaide’s Veterinary Diagnostic Laboratory (Roseworthy Campus, Roseworthy, Australia). Methods were utilized by a previously described method [28] where 150 µL swine antigen, and 0.5, 0.25, 0.125 and 0.063 mg/mL of purified swine IgG were used in place of the ovine antigen and purified ovine IgG, respectively. The colostrum and serum samples were diluted with PBS to a 1:160 and 1:100 dilution, respectively. Frozen-thawed piglet serum samples were analysed for total blood serum protein (%) using a digital hand-held refractometer (Atago: Model PAL-11S) following the manufacturer instructions. 

### 2.4. Statistical Analysis

Three control sows were removed from the trial. One due to savaging, a further one due to not eating and lastly due to abortion. These three sows were excluded from the statistical analysis. Therefore, the analysis is based on 21 control sows, 24 Low sows, and 24 High sows. During replicate 2, a severe heat event occurred affecting three sows, and data from these sows (*n* = one per treatment) has been excluded from the analysis post-heat-stress event. The software SPSS version 26 (IBM SPSS Statistics for Windows, v26.0. Armonk, NY, USA) was used for all analyses and results are expressed as mean ± standard error of the mean (SEM). A *p-*value < 0.05 was accepted as significant, and a *p-*value < 0.10 was considered a trend. For all analyses, non-significant fixed factors were removed from the model unless the factors were significant in a 2-way interaction. Sow data (melatonin and calcium plasma concentrations, sow live weight, P2 backfat, farrowing characteristics, colostrum % total solid, colostrum IgG concentration and litter characteristics) were analysed using a general linear model (for scale variables) or a multi-nominal logistic regression (for ordinal variables) with treatment (control, 0.16%; Low, 0.42%; High, 0.56%), parity, total born/suckled litter size (including fostered piglets) and replicates fitted as fixed factors. For piglet analyses, 113 piglets were removed to avoid ambiguity due to fostering outside of treatment. Therefore, the final number of piglets included in the analyses was 817 (control, *n* = 276; Low, *n* = 263; High, *n* = 278). The piglet survival was analysed using a Kaplan-Meier Log Rank (Mantel-Cox) test; causes of piglet mortality were analysed using a chi-squared analysis, with the remaining piglet data analysed using a General Linear Model. Treatment, replicate, parity, sex and total born/suckled litter size (including fostered piglets) were used as fixed factors and sow ID was fitted as a random factor. Time to stand and suck were log-transformed to provide a normal distribution of the data to determine significance, with the back-transformed means and SEM presented.

## 3. Results

### 3.1. Piglet Viability and Survival

The probability of piglet survival was significantly higher for both Low and High tryptophan treatments compared with the Control group (Figure 1; *p* < 0.001). The overall survival rate for control, Low and High tryptophan was 72.7 ± 2.7, 85.0 ± 2.2 and 83.0 ± 2.3%, respectively. Piglet viability measures were unaffected by treatment (Table 2). 

### 3.2. Piglet Weight and Weight Gain

Low and High tryptophan supplements had no effect on piglet weight or piglet weight gain at any time point compared with the control group (Table 3). There was also no effect on piglet weight when analysed as a repeated measure (*p* > 0.05). Low and High tryptophan had no effect on piglet weight or weight gain of low birth weight piglets (<1.1 kg) at any time point compared with the control treatment (*p* > 0.05). 

### 3.3. Mortality Causes

There tended (*p* = 0.059) to be fewer mortalities due to deformities within the High compared to the Low tryptophan treatment group (Table 4). There were no significant differences for the other causes of mortality. The mean mortality age, including stillbirths, was 1.27 ± 0.26 and 1.94 ± 0.4 days, excluding stillbirths.

### 3.4. Sow Characteristics

Sows received tryptophan for an average of 14.2 ± 0.2 days (7.2 ± 0.2 days prior to farrowing). At 21:00, High tryptophan sows had significantly lower melatonin plasma concentrations compared with low tryptophan sows; however, there was no difference at any other time point (Table 5). Tryptophan supplementation had no effect on sow calcium plasma concentrations (Table 5). 

Tryptophan had no effect on farrowing characteristics (Table 6) or litter size at weaning (9.82 ± 0.205; *p* > 0.05). Tryptophan also had no effect on sow weight or P2 backfat at farrowing house entry (285.8 ± 5.0 kg: 15.2 ± 0.6 mm), day 1 (261.0 ± 5.2 kg: 13.9 ± 0.6 mm) or day 17 (257.4 ± 4.8 kg: 12.5 ± 0.5 mm) of lactation (*p* > 0.05). Litter size and litter weights at 4 h, 24 h, 3 d, 7 d and 17 d were unaffected by treatment (*p* > 0.05) Gestation length was also unaffected by treatment (115 ± 0.1 days).

## 4. Discussion

Pre-weaning mortality is a significant economic and welfare issue affecting the pig industry [29]. In our study, both Low and High tryptophan supplementation increased the probability of piglet survival compared with the control group. Thus, the results support the hypothesis that tryptophan supplementation will increase piglet survival. Our findings are supported by previous evidence of a positive linear relationship between total tryptophan (0.19–0.31% per kg of feed) and piglet pre-weaning survival [11]. In contrast, other studies have reported no effect of increasing total tryptophan levels from 0.20 to 0.32% per kg of feed [12], or from 0.26 and 0.30% per kg of feed [30] on pre-weaning survival. This may reflect the low sample size used in the study of Miao et al. (2019) (*n* = 8 sows/treatment) [12]. It is also possible that the considerably smaller increase in dietary tryptophan (0.04%) utilised by Varvel (2019) compared to the current study (>0.1%) may have been too small to elicit a physiological change [30]. 

A numerical, but not significant, decrease in stillbirths was observed in sows receiving the Low tryptophan treatment, which is in partial agreement with previous evidence of significant reductions in stillbirths as tryptophan levels increased [11]. In contrast, it has previously been reported that when sows received a higher dose of tryptophan in their morning feed, there was a significant increase in stillbirths [13]. Although it is important to note that Xu et al. [13] supplemented tryptophan for the 30 days prior to farrowing, whereas the majority of tryptophan studies supplemented tryptophan upon entry of the farrowing house (6–11 days preceding farrowing) [10,11,12]. Thus, the increase in stillbirths reported by Xu et al. [13] may be attributed to the prolonged duration of supplementation, as opposed to the dose itself. Due to the varying results, further research is needed to conclusively determine the effects of tryptophan supplementation on the incidence of stillborn piglets.

Piglet birth weight is one of the major determinants of piglet survival as it impacts both the thermoregulatory ability and the growth rate of the piglet [31]. Consistent with previous studies in which sows received supplemental tryptophan for a similar duration prior to farrowing [12,30], we observed no effect of either the low or high tryptophan treatment on piglet birth weight. In contrast, birth weight was reduced in piglets born to sows that received tryptophan levels of 0.39% per kg of feed compared with 0.13% per kg of feed in the morning for the 30 days preceding parturition [13]. It can, therefore, be concluded that short periods of tryptophan supplementation (6–11 days) prior to farrowing have no effect on piglet birth weight. 

Piglet weight gain up-to-weaning and weight-at-weaning were also unaffected by tryptophan supplementation, which agrees with previous evidence [10,11,23,30,32] that increasing total tryptophan levels above 0.16% per kg of feed (the control level in the present study) does not significantly increase piglet weight or weight gain. The lack of an effect of tryptophan supplementation on piglet weight gain may be explained, at least in part, by the lack of an effect on piglet viability at birth. Piglet viability is a reflection of the trauma experienced during the birth process and determines the ability of the piglet to thrive and survive after birth. To the best of our knowledge, this is the first study to investigate the impact of maternal tryptophan supplementation on indices of birth trauma, piglet viability and vigour. Our data indicate that meconium staining (an indicator of oxygen deprivation during parturition), as well as time to stand and suck (indicators of piglet viability), were unaffected by tryptophan supplementation. Similarly, two common indicators of colostrum intake, blood glucose and serum IgG levels [33], were similar in piglets born to control and tryptophan supplemented sows, and piglet rectal temperature was also similar for all treatments. Taken together, these findings indicate that tryptophan supplementation had no effect on piglet viability; however, it is worth noting that the degree of meconium scoring was low in all piglets, which is indicative of a low level of parturition-induced hypoxia.

It was interesting that despite there being no difference in piglet viability or growth, that piglet survival was higher in both tryptophan treatments compared with the control. The three main factors responsible for piglet mortality are piglet factors (viability and birth weight), environmental factors and sow factors [31]. Since there was no difference in piglet factors, and all treatments were exposed to the same environmental conditions, it is likely the increased survival rate seen in this study may be due to sow factors. Tryptophan supplementation can reduce aggression and cortisol levels by increasing serotonin production in group-housed gestating sows and piglets post-weaning [22,34]. Furthermore, Condous and van Wettere (unpublished) showed that a total tryptophan level of 0.6% per kg of feed decreased sow plasma cortisol levels 2–4 h post farrowing and decreased the duration and incidence of sows standing post-farrowing which tended to reduce overlays. There was no difference in mortality causes between treatments in the current study, aside from High tryptophan, which tended to have fewer deformed piglets compared to Low tryptophan. However, this is unlikely to be a result of tryptophan supplementation since deformities, especially splay leg syndrome, which was the most prevalent deformity in the present study, are due to management and/or genetic factors [35]. However, in the present study, overlays were the leading cause of mortality for all treatment groups. Although not significant, the Low tryptophan group had the least overlays, followed by High tryptophan and then control. Thus, more research should be conducted on the effects of tryptophan supplementation on sow behaviour and serotonin production to understand how tryptophan affects piglet survival. Furthermore, serotonin production may not increase linearly with increasing tryptophan levels [21]; thus, the similar survival rates seen in both High and Low tryptophan may be attributed to High tryptophan having a plateauing effect on serotonin production. 

Tryptophan increases both serotonin and melatonin production through the hydroxylation pathway [36]. It has been previously demonstrated that tryptophan supplementation, through the effects of serotonin, increased calcium plasma concentrations in both cattle and rats, respectively, which can prevent hypocalcaemia [17,24]. Additionally, tryptophan supplementation increases melatonin production in ewes [18]. It has been demonstrated that increasing melatonin in fetal lambs increased umbilical blood flow supplying nutrients and oxygen, which increased fetal growth and reduced the effects of hypoxia [20]. Thus, it was anticipated that the effects of tryptophan supplementation in the studies mentioned above would also be seen in sows and their offspring. In the present study, tryptophan supplementation had no effect on either plasma calcium or melatonin production, refuting the hypothesis that tryptophan supplementation increases plasma calcium and melatonin concentration. This is likely due to the studies mentioned above using the hydroxylated version of tryptophan (5-hydroxytryptophan), which directly converts to serotonin and melatonin, whereas only 1% of the tryptophan used in the present study is converted to 5-hydroxytryptophan [36]. Thus, this warrants investigation into the effects of 5-hydroxytryptophan on both plasma melatonin and calcium concentrations in sows.

## 5. Conclusions

In conclusion, Low (0.42%) and High (0.56%) tryptophan supplementation increased the probability of piglet survival compared to the control group (0.16%). There was no difference in piglet viability and growth, which are both correlated with piglet survival; thus, more research into the effects of tryptophan on piglet survival, particularly through sow behaviour, is warranted. Furthermore, maternal tryptophan supplementation did not increase sow plasma melatonin and calcium concentrations; however, there is merit in further research investigating the effect of 5-hydroxytryptophan supplementation.

## Figures and Tables

**Figure 1 animals-11-02540-f001:**
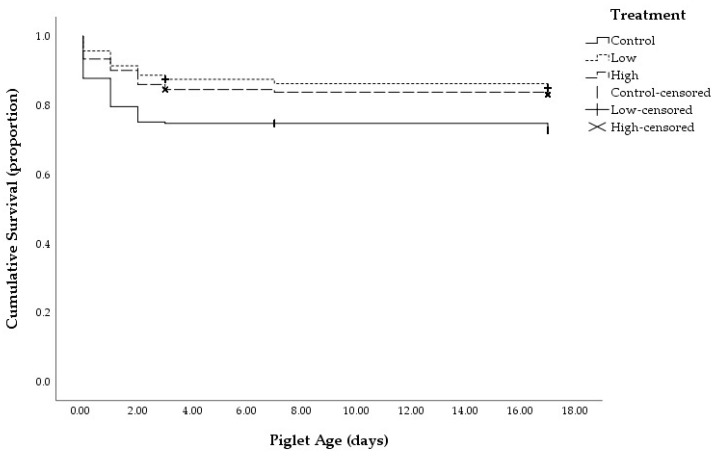
Piglet survival from birth to day 17 displayed as a Kaplan-Meier product limit estimates of survival plot of piglets born to sows in either the control (0.16%), Low (0.42%) or High (0.56%) treatment group.

**Table 1 animals-11-02540-t001:** The total amount of tryptophan (g) present in the control (0.16%), Low (0.42%) and High (0.56%) tryptophan diets at 7.2 ± 0.2 days prior to farrowing (PF) until day 7 of lactation.

Treatment	PF	Day of Lactation						
		1	2	3	4	5	6	7
Control	4.00	2.00	3.01	4.00	4.00	6.00	6.00	6.00
Low	10.50	5.25	7.52	10.50	10.50	15.75	15.75	15.75
High	14.00	7.00	10.53	14.00	14.00	21.00	21.00	21.00

**Table 2 animals-11-02540-t002:** The effect of tryptophan supplementation on piglet viability measures of piglets born to sows in either the control (0.16%), Low (0.42%) or High (0.56%) treatment group.

Piglet Viability Measures	*n*	Treatment			*p*-Value
		Control	Low	High	
Time to stand ^1^, min	252	4.2 ± 0.2	5.0 ± 0.1	3.7 ± 0.1	0.350
Time to suck ^1^, min	239	38.8 ± 0.2	26.0 ± 0.1	30.2 ± 0.1	0.102
Meconium stain score	314	1.9 ± 0.1	1.8 ± 0.1	1.8 ± 0.1	0.289
24 h blood glucose, mmol/L	175	5.4 ± 0.2	5.4 ± 0.2	5.7 ± 0.1	0.315
Serum IgG, mg/mL	165	48.4 ± 3.5	44.0 ± 3.2	46.1 ± 3.2	0.653
Serum total solids, %	165	6.2 ± 0.3	6.2 ± 0.3	6.0 ± 0.3	0.754
4 h rectal temperature, °C	629	37.9 ± 0.2	38.0 ± 0.1	37.9 ± 0.1	0.343
24 h rectal temperature °C	621	38.5 ± 0.1	38.5 ± 0.1	38.4 ± 0.1	0.666

Data presented as mean ± SEM, where *n* = total piglets in each measure. ^1^ Time to stand and suck from birth.

**Table 3 animals-11-02540-t003:** The effect of tryptophan supplementation on piglet weight, weight gain and average daily gain (ADG) from 4 h to weaning of piglets born to sows in either the control (0.16%), Low (0.42%) or High (0.56%) treatment group.

Piglet Weight (kg)	*n*	Treatment			*p*-Value
		Control	Low	High	
4 h	786	1.5 ± 0.0	1.5 ± 0.0	1.5 ± 0.0	0.815
24 h	653	1.6 ± 0.1	1.6 ± 0.5	1.7 ± 0.1	0.934
Day 3	636	1.9 ± 0.1	1.9 ± 0.1	2.0 ± 0.1	0.883
Day 7	619	2.8 ± 0.1	2.9 ± 0.1	2.9 ± 0.1	0.537
Day 17	598	5.4 ± 0.2	5.4 ± 0.2	5.4 ± 0.2	0.944
Piglet weight gain (g)					
4 h to 24 h	654	58.3 ± 14.8	64.1 ± 14.8	80.4 ± 15.1	0.436
24 h to day 3	630	180.2 ± 9.9	159.4 ± 8.7	155.0 ± 8.8	0.134
Day 3 to day 7	617	210.3 ± 10.4	205.8 ± 10.1	213.1 ± 10.1	0.867
Day 7 to day 17	597	261.6 ± 11.2	258.1 ± 10.8	259.3 ± 10.6	0.975
ADG (4 h to day 17; g/d)	598	228.0 ± 8.8	223.6 ± 8.6	226.4 ± 8.4	0.935

Data presented as mean ± SEM, where *n* = total piglets in each measure.

**Table 4 animals-11-02540-t004:** Piglet mortality causes of piglets born to sows in either the control (0.16%), Low (0.42%) or High (0.56%) treatment group.

Mortality Cause (%)	*n*	Treatment			*p-*Value
		Control	Low	High	
Overlay	60	45.3 (29)	34.2 (13)	36.0 (18)	0.448
Stillbirths	51	34.4 (22)	28.9 (11)	36.0 (18)	0.773
Low birthweight	15	7.8 (5)	7.9 (3)	14.0 (7)	0.489
Deformed	12	7.8 (5) ^xy^	15.8 (6) ^x^	2.0 (1) ^y^	0.059
Hypoxia	5	1.6 (1)	5.3 (2)	4.0 (2)	0.564
Exposure/starvation	1	0.0 (0)	2.6 (1)	0.0 (0)	0.221
Other	8	3.1 (2)	5.3 (2)	8.0 (4)	0.512

Data presented as mean ± SEM, where *n* = total piglets in each measure. Numbers in parentheses indicate total number of mortalities. ^xy^ Different superscripts within a row indicate *p* < 0.10.

**Table 5 animals-11-02540-t005:** Effect of tryptophan supplementation on sow plasma melatonin at 4.9 ± 0.2 days of lactation and calcium concentration from sows in either the control (0.16%), Low (0.42%) or High (0.56%) treatment group.

Melatonin, pg/mL	*n*	Treatment			*p*-Value
		Control	Low	High	
11:00	14	5.0 ± 1.6	5.3 ± 1.5	5.3 ± 1.5	0.987
16:00	14	4.0 ± 1.0	4.4 ± 1.0	3.4 ± 1.0	0.779
21:00	16	17.0 ± 2.0 ^ab^	21.0 ± 1.8 ^a^	11.2 ± 2.0 ^b^	0.011
01:00	14	17.7 ± 1.7	20.5 ± 1.7	14.8 ± 1.9	0.128
05:00	15	13.5 ± 2.3	16.5 ± 2.0	15.5 ± 2.5	0.622
Calcium, nmol/uL					
Day 113 of gestation	13	2.6 ± 0.1	2.7 ± 0.1	2.8 ± 0.1	0.198
Day 114 of gestation	11	2.8 ± 0.1	2.8 ± 0.0	2.8 ± 0.0	0.824
Day 1 of lactation	13	2.9 ± 0.1	2.7 ± 0.1	2.8 ± 0.1	0.507
Day 2 of lactation	14	3.0 ± 0.2	2.7 ± 0.2	2.8 ± 0.1	0.583

Data presented as mean ± SEM, where *n* = total sows in each measure. ^ab^ Different superscripts within a row indicate *p* < 0.05.

**Table 6 animals-11-02540-t006:** Effect of tryptophan supplementation on sow farrowing characteristics from sows in either the control (0.16%), Low (0.42%) or High (0.56%) treatment group.

Farrowing Characteristics	*n*	Treatment			*p*-Value
		Control	Low	High	
Total born, *n*	69	13.2 ± 0.8	11.6 ± 0.8	12.0 ± 0.6	0.571
Born alive, *n*	69	12.1 ± 0.7	11.1 ± 0.8	11.25 ± 0.6	0.939
Stillborn, *n*	69	1.1 ± 0.4	0.5 ± 0.2	0.8 ± 0.2	0.453
Mummified, *n*	69	0.5 ± 0.1	0.3 ± 0.1	0.5 ± 0.2	0.479
Stillborn ^1^, %	69	7.8 ± 2.0	3.7 ± 2.3	6.0 ± 2.3	0.311
Piglet birth interval, min	27	33.5 ± 6.9	30.7 ± 5.2	25.2 ± 5.2	0.524
Litters with low birthweight ^2^, %	69	14.6 ± 2.7	14.2 ± 2.7	13.5 ± 2.6	0.954
Colostrum total milk solids, %	35	25.5 ± 1.3	24.4 ± 1.1	24.8 ± 1.1	0.792
Colostrum IgG, mg/mL	37	53.5 ± 4.5	45.4 ± 3.9	44.8 ± 3.7	0.301

Data presented as mean ± SEM, where *n* = total sows in each measure. ^1^ The % of the litter that was stillborn. ^2^ The % of litters that contained piglets < 1.1 kg.

## Data Availability

The data is available upon request to the corresponding author.

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
