# Peer review of "Supplementary Tryptophan Fed to Sows Prior to and after Farrowing to Improve Piglet Growth and Survival"

_animals, 2021, doi:10.3390/ani11092540_

Round 1
Reviewer 1 Report
This manuscript by Munn and colleagues has determined the effects of the sows’ dietary supplementation of tryptophan on the piglet growth and survival. The experiment was well designed and data was well presented. This study finding that supplementary tryptophan fed to sows prior to and after farrowing could improve the growth and survival for the piglet. These findings will beneficial to the field. However, there are few issues should be resolved before the publication.
- The writing has many careless errors and needs to be seriously improved.
- Line 35, why not correct “ P = 0.011” to “ P < 0.011”? Please check the similar issues throughout the manuscript.
- Line 72, please correct “(<1.1 kg)” to “(< 1.1 kg)”. Please check the similar issues throughout the manuscript.
- Line 247-249, please move the “Data presented as mean ± SEM, where n = total piglets in each measure” to the toot note. Meanwhile, please specified the n=? for each treatment.
- Line 258, please correct “(P = 0.059)” to “(P = 0.059)”.
- Table 2, please correct “5.0 ±0.1” to “5.0 ± 0.1”. Please check the similar issues throughout the manuscript.
- Lines 255-257, please specify the P value used for the statistical significant changes in the footnote. Meanwhile, please specify the values meaning, such as mean ± SEM? in the foot note.
- Lines 262-263, the table 4 has similar issues to the upper tables.
- Table 6, what dose “% Litters with low birth weight”, “% Stillborn” this means?
- Please make use all the form of the tables within the three lines, and each line should be the same thickness.
Author Response
Dear Reviewer 1,
Thank you for taking the time to review this manuscript. Here are my responses below.
Comment 1: Thank you for noticing these errors, they have been corrected.
Comment 2: Thank you for your comment. The value ‘P = 0.011’ is displayed as such since that is the exact value, hence why it is not displayed as P < 0.05. The other authors of this paper prefer to keep the exact value.
Comment 3: Errors are fixed
Comment 4: Thank you for your comment. The information has been moved to the footnote. However, the other authors wish to not include each individual n value within each treatment group for each single measure in all tables as it would distract the reader from the main message. Other papers have not done this. If the Editor prefers to have an n value for each treatment group for every measure in each table, then I will add this. The maximum n values in for sows in each treatment is displayed in line 219 and for piglets it is displayed in line 234.
Comment 5: Corrected.
Comment 6: Corrected.
Comment 7: The P values used to determine statistical significance is described in the methods (line 224). It is also described in the footnote of tables with significant findings, if there are no significant findings then it is not included. Value meanings (mean ± SEM) have been specified in the statistical analysis section in the methods (line 223).
Comment 8: Corrected.
Comment 9: Footnotes have been added to provide definitions for these measures.
Comment 10: Corrected.
Thank you for your time.
Kind regards,
Amy Munn
Reviewer 2 Report
In general a carefully done study with conclusions that are well supported by the results. A few questions are indicated on the manuscript.

Author Response
Dear Reviewer 2,
Thank you for taking the time to review this manuscript. Here are my responses below.
Comment 1: The citations for this paragraph are all included in the paragraph below. The paragraph has been re-worded to make this clearer.
Comment 2: Sows were artificially inseminated with a terminal sire semen mix. This has now been included in the manuscript.
Comment 3: The values were the wrong way around and have been corrected, thank you for noticing this.
Comment 4: They were randomly selected. This has now been included in the manuscript.
Comment 5: Apologies, I’m not certain what this question is asking. If this is regarding why treatments groups are shown as control, low and high and as opposed to 0.16 %, 0.42 % and 0.56 %, I have added the quantitative versions to table/figure descriptions and in the conclusion.
Comment 6: Litter size at birth is displayed in table 6 (line 294) and litter size at weaning has now been added to the sow characteristics section (line 288).
Comment 7: I have re-analysed the results and conducted a repeated measures test, the results (no effect) are now included in line 259.
Comment 8: No sire effect because terminal mix semen was used.
Thank you for your time.
Kind regards,
Amy Munn.